# *This* and *that* in depression: Cross-linguistic semantic effects

Line Kruse[1]*, Roberta Rocca[2], Emanuela Todisco[3], Cordula Vesper[1], Peter Thestrup Waade[4,5], Mikkel Wallentin[1,4,6]

**1** Department of Linguistics, Cognitive Science and Semiotics, Aarhus University, Aarhus, Denmark, **2** Department of Culture, Cognition, and Computation, Aarhus University, Aarhus, Denmark, **3** Universidad de Sevilla, Departamento de Lengua Española, Lingüística y Teoría de la Literatura, Sevilla, Spain, **4** Interacting Minds Centre, Aarhus University, Aarhus, Denmark, **5** Wellcome Centre for Human Neuroimaging, University College London, London, England, **6** Center of Functionally Integrative Neuroscience (CFIN), Aarhus University, Aarhus, Denmark

* linekruse@cc.au.dk

**Data availability statement:** Supplementary Materials, data and scripts for analyses are available at the Open Science Framework: https://osf.io/eq3j5/.

## Abstract

Demonstratives (in English "this" and "that") are pivotal to human communication, facilitating joint attention and the establishment of a common ground of reference. All languages have at least two forms, typically distinguishing proximal from distal space, where "space" is defined by a range of context-dependent physical, psychological, social and referent-intrinsic factors. Recent work based on the Demonstrative Choice Task (DCT) has indicated, that in the absence of a guiding context, demonstrative reference may capture an experienced or emotional proximity to the referent concepts, and that semantic differences in responses allow implicit inferences on individual differences in psychological space related to depression. The present paper investigated the extent to which these patterns generalize across languages, including German, Spanish, Italian, Russian, Chinese and Tagalog Filipino samples. DCT-based classification models robustly outperformed baseline models in all languages except for Chinese, and showed similar semantic patterns as observed for English. Particularly negative emotion dimensions were consistently among the most important features across models, for which choice of the proximal demonstrative form was more frequent in the depression group than the control group. The opposite pattern was observed for positive emotion features, however, these effects were more variable across languages. Results suggest that simple lexical choices in the DCT capture semantic differences in experiential states related to depression, and may be used to map individuals along a multidimensional semantic space, potentially providing novel insights into individual differences in disorder states and etiology.

## Introduction

Language and cognition are strongly intertwined, influencing and reflecting each other in bidirectional manners [1–5]. Differences in mental state such as mood or stress have been

**Funding:** This work was supported by the Aarhus University Research Foundation (AUFF.E-2020-9-3 to MW), which funded LK. The funders (https://auff.au.dk/en/) had no role in study design, data collection and analysis, decision to publish, or preparation of the manuscript. The authors have no competing interests to declare.

**Competing interests:** The authors have declared that no competing interests exist.

found to influence neural responses to linguistic information [6–8], and individual differences in personality traits and mood can be detected solely from produced text [9–12]. The depression literature is challenged by the permeating heterogeneity in symptom profiles and disorder trajectories [13,14], as well as extensive overlap in diagnostic criteria with other diagnoses [15]. This indicates that there may be important individual differences in disorder states beyond what is expressed in diagnostic symptoms. Language may be a useful tool to investigate the content of disorder states, providing a rich semantic feature space across which individuals can be mapped. Such semantic characteristics may provide more fine-grained information on the states underlying or contributing to diagnostic symptoms at the individual level.

Spatial demonstratives (in English "this" and "that") are a prime example of the strong link between language and cognition. Across languages, demonstratives are among the most frequent words in the adult lexicon [16] and among the first words learned during language acquisition [17,18]. All languages appear to have at least two demonstratives, corresponding to the English proximal ("this") and distal ("that") forms [19]. Demonstratives are deictic expressions, carrying very little meaning in themselves, but serving a pivotal role in establishing joint attention and common ground during natural interaction [20]. Proximal and distal forms typically distinguish referents in near and far space, where "space" is largely context-dependent and can be physical or conceptual. Peeters and colleagues [21] argued that three types of factors influence demonstrative use and comprehension in flexible and dynamic ways; physical factors pertaining to the external physical context (e.g. physical distance/location, visibility, position of the addressee etc.), psychological factors related to the cognitive status of the referent assumed by interlocutors (e.g. emotions/attitudes towards the referent, shared experience, etc.) and referent-intrinsic factors (e.g. ownership and familiarity). As such, it is argued that demonstrative reference is a product of conceptual processes [19], and their spatial meaning is determined by the relative proximity between speaker and referent in a conceptual space, influenced by several physical and psychological factors. Here, we address the extent to which demonstrative reference in the Demonstrative Choice Task can be used to infer individual differences in conceptual space related to depression across 7 languages.

## Demonstrative Choice Task (DCT)

In normal communication, the above factors, i.e. relative proximity to the referent, relative position of the interlocutor, or shared familiarity with the referent, act as contextual anchors guiding the choice of demonstrative depending on the communicative context or goal. For instance, a speaker may use the proximal demonstrative ("is *this* my cup?") to distinguish it from another cup slightly further away and reduce ambiguity about which cup is being referred to. The Demonstrative Choice Task (DCT) [22] assesses demonstrative reference in a setting with no external contextual anchors to guide the proximal/distal distinction. In the DCT, participants are repeatedly presented with one word in isolation, and are asked to match it with either the proximal or distal demonstrative form (in the case of English), based on their immediate preference (e.g., "this book" or "that book") (Fig 1). In this case, demonstrative choice is purely endophoric, and the conceptual space in which the choice is grounded is confined to psychological and referent-intrinsic factors. The task has been used to study the influence of semantics on demonstrative reference. DCT studies in English, Danish and Italian [23] have shown that choice of demonstratives for a sequence of nouns were highly consistent across subjects and structured by the semantic properties of the referent objects, such as animacy, manipulability/size and harmfulness. Using a broader semantic space, DCT studies

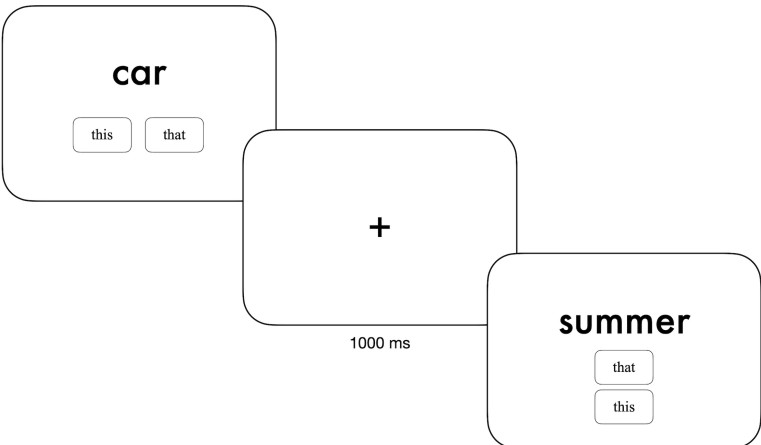

**Fig 1. Demonstrative choice task.** Illustration of two consecutive trials in the DCT, separated by a fixation cross presented for 1000 ms. In each trial, the participant is to select either the proximal ("this") or distal ("that") demonstrative form for the given word, based on their immediate preference. Responses were made with a mouse click. The two response buttons alternated between horizontal and vertical configurations, and the button labels (*this* or *that* were assigned at random on each trial.

in English [22], Spanish [24] and Catalan [25] found that nouns of high manipulability, positive valence, and related to "self" elicited higher proportions of proximal demonstrative forms, while items associated with negative valence, harmfulness, and low manipulability elicited higher proportions of distal demonstrative forms. These results replicated across languages of both two- and three-demonstrative-system languages (in casu Spanish and Catalan), and indicate that, with no external contextual anchors, demonstrative reference reflects an experienced or psychological proximity of the speaker to the referent. Leveraging on the spatial information inherent in demonstrative reference, the DCT may be a useful tool to investigate and map individual differences in conceptual space, such as maladaptive states associated with depression.

Depression is a diagnosis associated with exceptional heterogeneity in symptom profiles [13,14], disorder trajectories [26] and treatment efficiency [27]. Further, it is one of the DSM-IV diagnoses exhibiting highest comorbidity and largest overlap in diagnostic criteria with other diagnoses [15]. Identifying means to assess characteristics of depressive states beyond diagnostic symptoms may facilitate better understanding of individual disorder states that can guide therapeutic and treatment protocols. Several studies have provided evidence that differences in language production and processing are associated with depression. These include increased use of negative emotion words and self-referential language [10,28,29], altered cognitive processing of emotional information [8,30] and altered limbic activation for emotional linguistic information [6,7,31]. Further, depression is characterized by increased self-focused attention and maladaptive distortions in the experiential and narrative self [32, 33], often accompanied by rumination, the tendency to engage in repetitive negative thought patterns or self-talk [34,35]. If behavior in the DCT reflects self-referential experienced proximity to the referents, it may be used to infer such depressive states and allow inference on the semantic domains driving these, such as valence or fear. In a recent study based on a 290-item DCT we showed that individuals with high depression symptom scores (PHQ-9 sum score $\geq 10$) could be classified based on behavior on the DCT across two independent samples

(n = 775 and n = 879) [36]. The strongest predictive items of depression included higher tendency to choose proximal demonstratives for highly negatively valenced items, such as *pain*, *denial*, *loner*, *darkness* and *emptiness*, compared to individuals with low depression symptom scores. These findings supports the hypothesis that demonstrative reference in the DCT reflects psychological proximity between the mental/emotional experience of the participant and the referent, recovering semantic differences in mental states previously related to depression symptoms.

## Present study

There are several factors on which the findings reported in the English study may depend. First, there are considerable differences in demonstrative systems across languages, mainly in the number of demonstrative terms available, but also in the spatial information and deictic content inherent in the different demonstrative forms [21]. While many languages have a two-term demonstrative system, several languages have three-term systems or more. Further, while demonstratives of European languages mainly distinguishes relative distance (physical or conceptual), some languages have demonstrative forms directly conveying non-distance properties, such as geocentric location, elevation, direction or visibility to the speakers [19]. Such differences in demonstrative systems and deictic content may cause different choice patterns in the DCT related to depression. However, there may also be important cultural differences in depressive experience, perception and states. This could imply that the semantic features most predictive of depression in English samples may be different in other languages, albeit sharing similar demonstrative systems. The present study extends previous work on the relationship between demonstrative reference in the DCT and depression symptoms, addressing the extent to which these findings generalize across other European languages (German, Spanish, Italian and Russian) to languages of different families (Filipino and Chinese), across two-term (English, German, Italian, Russian, and Chinese) and three-term (Spanish and Filipino) demonstrative systems, and across cultures (North/South European, Slavic, and East/South-East Asian). Additionally, we extend previous work in addressing whether depression can be classified from semantic profiles of individuals, constructed from DCT behavior. This approach moves beyond word-level inference and allows more rigorous assessment of general semantic effects driving group differences in behavior.

First, a replication of the analysis performed in English was conducted in the six new languages, using logistic regression to classify individuals with high and low depression symptoms scores based on word-level responses in the DCT. Second, depression classification was conducted based on semantic profiles of participants, where each subject was assigned a score on each of the 65 semantic Binder features [37], aggregated across trials in the task. Higher scores on a feature indicated higher proportions of proximal demonstrative choices for items scoring high on this feature, while low scores indicated higher proportion of distal demonstrative choices across items scoring high on this feature. For analyses based on semantic profiles we included the English sample (from [36]) in addition to the six new languages, to assess shared and language-specific effects across all 7 languages. As the sample sizes across languages differed substantially, and were relatively small compared to those of previous work, we performed classification analyses on both the full sample, including all 7 languages (yielding a large sample size), and on each language sample individually. This allowed us to investigate classification performance in a large cross-linguistic sample, where important features may reflect cognitive patterns associated with depression that are shared across languages. Contrary, the individual analyses for each language could identify potential differences in

classification performance between languages, and allow comparison of language-specific semantic effects.

## Methods

The project was approved by the Institutional Review Board at Aarhus University. All participants provided written consent prior to participating in the study. Recruitment of participants was conducted in the period 11/03/2024 to 30/05/2024.

### Sample descriptives

We conducted an online 290-item DCT study in German, Italian, Spanish, Chinese, Russian and Filipino. Participants were recruited via the online platform Prolific.co (https://www.prolific.co), and were native speakers of the respective language. The aim was to recruit 500 participants for all language groups. This goal, however, was only partially obtained within our time-frame for sampling. 500 participants were recruited for the German, Spanish and Italian studies. 335 individuals participated in the Chinese study, 317 individuals participated in the Russian study and 230 individuals participated in the Filipino study. Participants were excluded from analyses if they fulfilled one of three conditions indicating low effort responses; 1) reaction times below 300 ms in more than 10% of the trials, 2) response-button entropy below 0.80, indicating a consistent response pattern irrespective of the stimuli, and 3) more than 3 of 15 failed attention checks (see below). Additionally, a total of 23 participants were excluded due to missing data caused by server errors. The English sample collected in Kruse et al. (2024) [36] was additionally included in the second part of the analysis, for comparison across languages. Participant demographics across language samples are reported in Table 1.

**Table 1. Sample descriptives by language.**

| Sample | N Participants | | N Excluded | | Gender Distribution | | | |
|---|---|---|---|---|---|---|---|---|
| | Collected | Included | Low-Effort | Missing Data | Female | Male | Non-binary | Other |
| English | 2068 | 1654 | 58 | 356 | 762 | 873 | 16 | 3 |
| German | 500 | 488 | 8 | 4 | 238 | 240 | 8 | 2 |
| Italian | 500 | 462 | 35 | 3 | 219 | 237 | 6 | 0 |
| Spanish | 500 | 492 | 1 | 7 | 241 | 239 | 11 | 1 |
| Chinese | 335 | 279 | 52 | 4 | 198 | 79 | 2 | 0 |
| Filipino | 230 | 199 | 27 | 4 | 134 | 56 | 5 | 4 |
| Russian | 317 | 307 | 9 | 1 | 211 | 90 | 6 | 0 |
| Total | 4450 | 3881 | 190 | 379 | 2003 | 1814 | 54 | 10 |
| Sample | Age Distribution | | | PHQ9 Score Distribution | | | | |
| | Mean (SD) | Min | Max | N Depression | N Contol | Median (SD) | Min | Max |
| English | 40.95 (13.68) | 18 | 89 | 534 | 1120 | 6 (6.04) | 0 | 26 |
| German | 34.4 (11.6) | 19 | 74 | 185 | 303 | 7.5 (5.59) | 0 | 27 |
| Italian | 33.4 (10.6) | 20 | 72 | 196 | 266 | 8 (6.02) | 0 | 27 |
| Spanish | 33.4 (10.6) | 20 | 72 | 170 | 322 | 7 (6.09) | 0 | 27 |
| Chinese | 31.4 (9.2) | 18 | 62 | 59 | 220 | 6 (5.25) | 0 | 27 |
| Filipino | 33.4 (0.8) | 18 | 68 | 75 | 124 | 7 (5.71) | 0 | 22 |
| Russian | 33.5 (10.9) | 19 | 79 | 118 | 189 | 8 (5.23) | 0 | 26 |
| Total | 36.62 (12.50) | 18 | 89 | 1337 | 2544 | 7 (5.89) | 0 | 27 |

## Materials

**Demonstrative Choice Task (DCT).** Participants performed a 290-item Demonstrative Choice Task, identical to the one used in Kruse and colleagues (2024) [36]. One noun was presented in the center of the screen for each trial, and participants were asked to match it with either the proximal ("this") or distal ("that") spatial demonstrative forms, presented below the target noun (Fig 1). For Spanish and Filipino, participants were to match each noun with one of the three spatial demonstrative forms present in the language (Table 2). Since spatial demonstratives in German and Tagalog Filipino are *adverbs*, rather than *adjectives*, which is the case for English, Spanish, Italian, Chinese and Russian, their use requires the addition of a determiner in accordance with the noun gender (for details on the demonstrative systems of each of the six included languages and translation of the task, see S1 Supplementary Experimental Procedures). Each noun was presented until a response was made. Response buttons changed labels ("this" or "that") at random, and alternated between two spatial configurations (vertical or horizontal). Responses of two-demonstrative languages were recoded as 1 (proximal) and –1 (distal). Responses of three-demonstrative languages were recoded as 1 (proximal), 0 (medial) and –1 (distal).

**Depression symptom scores.** Depression symptom scores were assessed using the Patient Health Questionnaire 9-item (PHQ-9) [38] rating the severity of 9 symptoms on a scale from 0-3. A PHQ9 sum score $\geq$ 10, indicating moderate to severe depression, was used as threshold for classifying participants into either "control" or "depression" group. Groups were coded as 0 (control) and 1 (depression).

**Binder ratings of DCT items.** The Binder dataset, which formed the basis for some of our previous investigations of demonstrative choice [22,24,25], contains human ratings of 535 words across 65 semantic dimensions, thought to reflect basic neurocognitive components of meaning [37]. The word list used in the present version of the DCT was adapted to better reflect semantics related to mood [36] and thus includes items not originally part of

**Table 2. Examples of translated DCT trials in German, Spanish, Italian, Russian, Filipino and Chinese.** "Options" refer to the response options presented for the given language. "Prox", "dist" and "med" indicate proximal, distal and medial demonstratives, respectively.

| English | | | German | | | Italian | | | Spanish | | | |
|---|---|---|---|---|---|---|---|---|---|---|---|---|
| **Stimulus** | **Options** | | **Stimulus** | **Options** | | **Stimulus** | **Options** | | **Stimulus** | **Options** | | |
| | Prox | Dist | | Prox | Dist | | Prox | Dist | | Prox | Med | Dist |
| family | this | that | die Familie | hier | da | famiglia | questa | quella | familia | esta | esa | aquella |
| world | this | that | die Welt | hier | da | mondo | questo | quel | mundo | este | ese | aquel |
| problem | this | that | das Problem | hier | da | problema | questo | quel | problema | este | ese | aquel |
| dress | this | that | das Kleid | hier | da | vestito | questo | quel | vestido | este | ese | aquel |
| pain | this | that | der Schmerz | hier | da | dolore | questo | quel | dolor | este | ese | aquel |
| company | this | that | das Unternehmen | hier | da | impresa | quest' | quell' | empresa | esta | esa | aquella |
| criminal | this | that | der Verbrecher | hier | da | criminale | questo | quel | criminal | este/a | ese/a | aquel/la |
| jealousy | this | that | die Eifersucht | hier | da | gelosia | questa | quella | celos | estos | esos | aquellos |

| English | | | Russian | | | Chinese | | | Tagalog | | | |
|---|---|---|---|---|---|---|---|---|---|---|---|---|
| **Stimulus** | **Options** | | **Stimulus** | **Options** | | **Stimulus** | **Options** | | **Stimulus** | **Options** | | |
| | Prox | Dist | | Prox | Dist | | Prox | Dist | | Prox | Med | Dist |
| family | this | that | семья́ | эта | та | 家庭 | 这个 | 这个 | ang pamilyang | ito | iyan | iyon |
| world | this | that | мир | этот | тот | 世界 | 这个 | 那个 | ang mundong | ito | iyan | yion |
| problem | this | that | пробле́ма | эта | та | 问题 | 这个 | 那个 | ang problemang | ito | iyan | iyon |
| dress | this | that | пла́тье | это | то | 连衣裙 | 这件 | 那件 | ang damit na | ito | iyan | iyon |
| pain | this | that | боль | эта | та | 痛苦 | 这个 | 那个 | ang sakit na | ito | iyan | iyon |
| company | this | that | компа́ния | эта | та | 公司 | 这个 | 那个 | ang kompanyang | ito | iyan | iyon |
| criminal | this | that | престу́пник | этот | тот | 罪犯 | 这位 | 那位 | ang kriminal na | ito | iyan | iyon |
| jealousy | this | that | ре́вность | эта | та | 妒忌 | 这个 | 那个 | ang selos na | ito | iyan | iyon |

the Binder wordset with associated ratings. Ratings on Binder features for all 290 items in the task were obtained by learning a mapping between the 300-dimensional Glove semantic vectors [39] and the 65-dimensional Binder space, following the procedure described in Turton and Smith (2020) [40]. Six models were trained on the Binder dataset (excluding the words included in the DCT item set), to predict word scores on each of the 65 dimensions from the respective Glove vectors (for details see S3 Supplementary Experimental Procedures). The model explaining most of the variance across all 65 features (mean R2), a 3-layer neural network, was used to predict Binder scores for the 290 DCT items, yielding a 65-dimensional semantic vector for each item in the task (see DCT item scores on each feature in S3 Supplementary Experimental Procedures). Note that the semantic Binder scores included in the present study are based on human ratings of English words.

## Replication of analysis from English data

To assess whether the results observed in our previous study with English speaking participants [36] generalized to other languages, we first performed a replication of the previous analysis. This involved predicting the outcome group from principal component (PC) representations of the raw response matrix (participants x words). PCA was performed to reduce dimensionality and multicollinearity of the word-level responses (290 predictors) for statistical analysis. The replication analysis included all 6 new languages. Two DCT-based models were evaluated including 1) DCT responses as input (mDCT) and 2) DCT responses, gender, and age as input (mDCT+GenderAge). Performance of these were compared to two baseline models; 1) only gender and age as input (mGenderAge) and 2) a model trained on randomly shuffled outcome labels (mRandomBaseline). All models were trained on 70% of the data and evaluated on 30% of the data, stratified by outcome label and language sample. Data was downsampled to balance class prevalence, yielding a chance performance accuracy at 0.5. For details on this replication analysis, see S2 Supplementary Experimental Procedures.

## Classification by semantic profiles

The original English study [36] and the replication analysis in the 6 new languages used item-level responses as input for the depression classification. Results indicated specific DCT words that were associated with robust differences in demonstrative choice behavior predictive of depression symptom severity. If these patterns indeed reflect general semantic differences, semantically similar words should elicit similar effects. Contrary, the observed effects may pertain to those specific words, rather than general semantic differences. To investigate the extent to which depression related differences in demonstrative choice reflect differences along generalized semantic dimensions, beyond individual words, we used subject-specific semantic scores aggregated across choices for all items as input to the classification model. That is, for each subject, a semantic profile was computed, based on their responses across all items in the DCT. If the differences between the depression and control groups reflects a word-effect, rather than a semantic effect, aggregated semantic input features (across words) should yield little effect in the classification model and model performance should be poor. Contrary, if the differences in responses are consistent along specific semantic features, then these features should yield additive predictive effects in the classification model. In the latter case, this would suggest that there are semantic dimensions for which demonstrative reference are predictive of depression, potentially indicating semantic characteristics of an altered psychological space.

**Input features.** For each participant a semantic vector of 65 (Binder) dimensions was computed , based on responses on the 290 unique DCT items. Semantic vectors were computed by matrix multiplication of the Binder matrix [words x semantic features] and the response matrix [subjects x words], yielding a [subject x semantic features] matrix. Thus, each participant was assigned a score on each semantic feature, reflecting the aggregated effect of demonstrative choices (positive/proximal, zero/medial, negative/distal) for the respective semantic feature across all 290 items. For instance, the three words "party", "car", and "shame" score 5.9, 2, and 2.5 on the semantic dimension *social*, respectively. On the semantic dimension *sad* they score 0.03, 0.1, and 4.8, respectively. Say subject A responded with "this" (1), "that" (−1) and "that" (−1) to the three words, respectively. The mean score for subject A on the semantic feature *social* would then be ((1*5.9)+(−1*2)+(−1*2.5))/3 = 0.5. The mean score for subject A on the semantic feature *sad* would be ((1*0.03)+(−1*0.1)+(−1*4.8))/3 = −1.6. This was computed across all 290 words in the task for all 65 semantic features, yielding a 65-dimensional semantic vector for each subject based on their demonstrative choices. Higher scores indicated more proximal responses to words scoring high on this feature, while lower scores indicated more distal responses to words scoring high on this feature. Scores approximating 0 indicate no difference in the frequency of proximal versus distal demonstrative responses on the given feature. Note that this procedure effectively ignores the medial responses in the Spanish and Filipino cases, and only compares the proximal/distal distinction across all 7 languages (including English). For comparison, the analysis was also performed contrasting *proximal* vs *medial+distal* and contrasting *proximal+medial* vs *distal* (see S4 Supplementary Experimental Procedures). Scores on all features were approximately Gaussian distributed across subjects, however, with a long positive tail for some features in the Spanish, Italian, Filipino and Chinese samples (S1 Appendix). The 65-dimensional semantic vectors were used as input to classification models predicting outcome group (depression vs. control).

**Classification model comparison.** Some of the 65 semantic features are highly correlated across words in the DCT, violating the assumption of no multicollinearity. To maintain all 65 semantic features in the analysis, we trained and evaluated five different models, each handling multicollinearity in different ways including regularization and/or penalization terms. This allowed data-driven assessment of the importance and effect of all 65 features. The compared models included a Ridge Regression, k-Nearest Neighbors, Decision Tree Ensemble (xgboost), a 2-layer (128 units, 64 units) and a 3-layer (128 units, 64 units, 32 units) Neural Network model. All hidden layers of the NN models used the *relu* activation function, while the output layer used the *sigmoid* activation function. All models were trained with hyperparameter tuning on a validation set stratified by outcome group and language sample. Binder scores were scaled prior to training, and the data was downsampled to balance class prevalence, yielding a chance performance accuracy at 0.5. The training set comprised 70% of the data and the hold-out test set comprised 30% of the data. 20% of the training sample was used as validation set for hyperparameter tuning. Hyperparameter tuning of the ridge model included *learning rate*, the kNN model was tuned on *n neighbors*, *leaf size*, and *weights*, tuning parameters for the Decision Tree Ensemble model included *n estimators*, *learning rate*, *max depth*, *alpha* (ridge regularization strength) and *lambda* (lasso regularization strength) and the neural networks were tuned on *batch size* and *n epochs*. The models were retrained on the full training set following hyperparmeter tuning, and performance was evaluated on F1 and ROC-AUC scores on the hold-out test set.

Importantly, the language distribution in the full sample was imbalanced. Hence, while the model may perform well in general, it could be the case that the model was mainly learning effects based on, for instance, the larger English sample and thus performed very poorly on

languages with fewer samples (Russian, Chinese, and Filipino). To assess out-of-sample performance differences between languages, the best model was additionally evaluated on each language-specific subset of the hold-out test set. Performance of the model was compared to a baseline model, trained on randomly shuffled versions of the outcome variable. F1 and ROC-AUC metrics on each test-set indicate classification performance across classes. Precision metrics indicate the fraction of cases classified as depression that were true cases of depression, while recall metrics indicate the fraction of positive cases (depression) that were indeed classified as depression by the model.

Predictive effects of semantic features were computed from the best model using SHAP values. The *shap* library [41] computes local Shapley values, quantifying the individual contribution (and direction) of each feature to the model prediction at each sample (prediction).

**Classification on individual language samples.** While a cross-linguistic model identify potential patterns shared between language samples, there may be important differences between languages not captured in such a model. To assess language-specific classification performance based on the DCT and potential differences in semantic patterns of responses, the analysis was additionally conducted in each language sample individually, following the same procedure as described for the cross-linguistic model. Thus, the model was hyperparameter optimized on each language sample individually and performance on the test set was compared to a language-specific random baseline model. SHAP values were computed for each language model allowing comparison of feature effects across samples.

**Robustness tests.** Previous results on the Demonstrative Choice Task in English [36] has shown that while depression classification models based on DCT behavior robustly perform better than baseline models, the exact performance estimates varies depending on the train- and test sampling, particularly with smaller sample sizes. Further, while multicollinearity does not affect performance of Decision Tree Ensemble models, it can affect the estimated contribution of features. In cases where two features are highly correlated and assigned a high decision weight by the model, one of them will be selected at random for the decision, resulting in high importance for the selected feature and low importance for the unselected feature, despite the two being highly correlated. To assess robustness of model performance against random variations in the data, bootstrapped robustness test was performed. Each model (full cross-linguistic model and language-specific models) was trained and evaluated 1000 times, each time sampling a new training and test set. This facilitated assessment of the variability of model performance and feature importance estimates against random variations in data sampling.

## Results

### Descriptive results

The mean PHQ-9 sum score across all samples was 7 (SD = 5.89, min = 0, max = 27), with lowest median scores in the Chinese and English sample (median=6) and highest scores in the Russian and Italian samples (median=8) (Table 1). 34.45% of all participants were classified as depression cases (PHQ-9 sum $\geq$ 10), while 65.55% of participants were classified as control cases. The proportion of depression cases ranged from 21.15% (Chinese) to 42.40% (Italian) (S1 Fig). The control group was equally gender distributed (F=50%, M=50%), while the depression group had a larger proportion of females (58%) compared to males (42%). All language samples had larger proportions of females compared to males in the depression group, with the largest difference in the Chinese depression group (F=83.05%) and the smallest difference in the Italian depression group (F=50.26%) (S2 Fig). The distribution of demonstrative choices was 52% for proximal, 42% for distal, and 7% for medial (only present in two

languages) forms (S3 Fig). The demonstrative choice distribution in the English sample was 47.4% for proximal and 52% for distal forms, in the German sample 49.2% for proximal and 50.8% for distal forms, in the Italian sample 68.4% for proximal and 31.6% for distal forms, in the Spanish sample 36.9% for proximal, 40.2% for medial, and 22.9% for distal forms, in the Chinese sample 61.1% for proximal and 38.9% for distal forms, in the Russian sample 70.2% for proximal and 29.8% for distal forms, and in the Filipino sample 46.4% for proximal, 27.6% for medial, and 26% for distal forms (S4 Fig). There were no differences in demonstrative choice distribution between depression and control group for any language sample (S5 Fig and S6 Fig).

## Replication analysis: Classification from DCT responses

The replication analysis used PC representations of DCT responses to classify depression in a sample including German, Spanish, Italian, Russian, Chinese and Filipino data. The analysis yielded similar results as those reported in Kruse et al. 2024 [36], and showed that depression could be classified significantly above chance based on DCT responses (S2 Supplementary Experimental Procedures). Similar to our previous study, the mDCT+GenderAge model performed best with mean accuracy of 0.62 across bootstraps (SD = 0.02. 95% CI = [0.61,0.62]), while the mDCT showed a mean accuracy of 0.58 (SD = 0.02), 95% CI = [0.57,0.58]). Only the mDCT+GenderAge model performed better than the baseline mGenderAge model which had a mean accuracy of 0.59 (SD = 0.02, 95% CI = [0.586,0.588]), while both DCT models outperformed the mRandomBaseline, which did not perform better than chance level. The 10 DCT words most predictive of depression were *shit*, *poverty*, *darkness*, *hell*, *emptiness*, *victim*, *jealousy*, *hunger*, *stupidity*, and *complaint*, in both the mDCT and mDCT+GenderAge models (S2 Supplementary Experimental Procedures). The top words predicting towards the control group included *sport*, *sense*, *office*, *team*, *era* and *hotel*, in both the mDCT and mDCT+GenderAge models.

## Classification by semantic profiles

**Cross-language classification.** The Decision Tree Ensemble model performed best on the hold-out test sample (F1 = 0.64, precision = 0.65, recall = 0.64, ROC-AUC = 0.65), followed by the ridge model (F1 = 0.60, precision = 0.59, recall = 0.62, ROC-AUC = 0.59), and the kNN model (F1 = 0.58, precision = 0.64, recall = 0.53, ROC-AUC = 0.62) (Fig 2). The two neural networks performed worst, predicting solely the majority class (F1 = 0.51, ROC-AUC = 0.59).

The selected hyperparameters of the best cross-linguistic model were *lr* = 0.1, *max depth*=9, *n estimators*=20, *alpha* = 0.01, and *lasso* = 0.001 (S1 Table). The model was additionally evaluated on each language-specific subset of the test sample and exhibited F1 scores above 0.60 for all language samples except for Chinese where performance was below chance (F1 = 0.4) (Table 3). Best performance was observed for the Italian sample (F1 = 0.66), followed by the English sample (F1 = 0.65) and the Russian sample (F1 = 0.64). While the model generalized well to the test set and performed best of all models, performance on the training set (cross-linguistic) was exceptionally high (F1 = 0.98) indicating a tendency for the model to overfit.

Bootstrapped robustness tests indicated that model performance was more sensitive to random variation in the training data for smaller samples, and more robust against random variations for larger samples. The model exhibited a median F1 score across bootstraps above 0.6 for the full sample, the Italian, Spanish, German, Filipino and English sample (Fig 3A). Median performance for the Russian sample was just below 0.6, while median performance

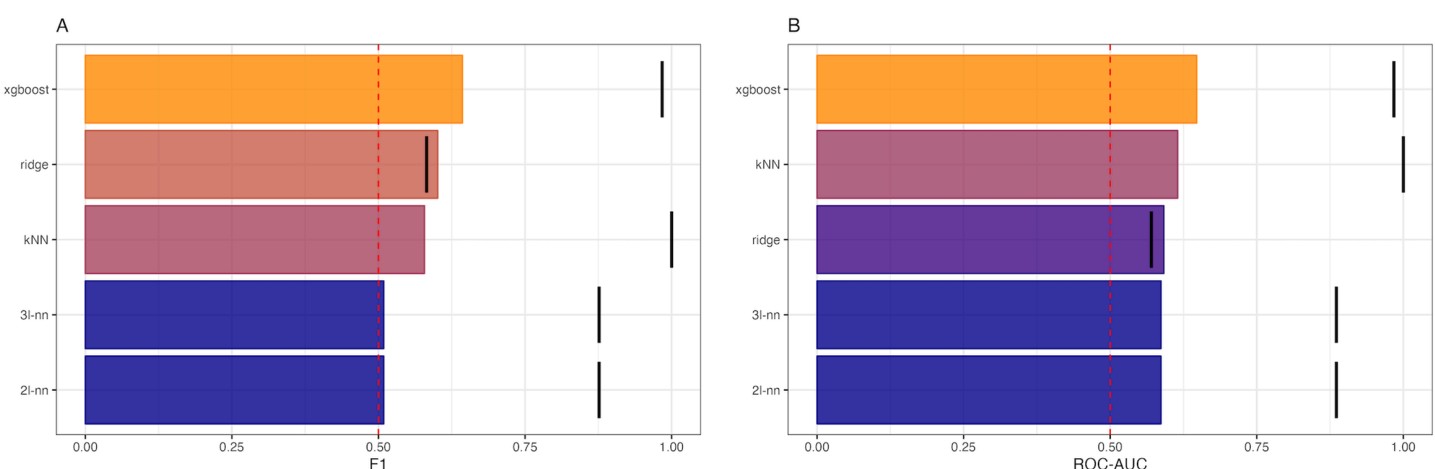

**Fig 2. Classification performance model comparison.** Performance of the five classification models on training set (black lines) and test set (colored bars) evaluated on A) F1 scores and B) ROC-AUC scores.

Table 3. Cross-linguistic model performance on full sample (train and test) and language-specific test sets.

| Evaluation | F1 | ROC-AUC | Precision | Recall |
|---|---|---|---|---|
| Full Train | 0.98 | 0.98 | 0.99 | 0.98 |
| Full Test | 0.64 | 0.65 | 0.65 | 0.64 |
| German | 0.62 | 0.65 | 0.68 | 0.57 |
| Spanish | 0.63 | 0.64 | 0.65 | 0.61 |
| Italian | 0.66 | 0.66 | 0.67 | 0.64 |
| English | 0.65 | 0.65 | 0.65 | 0.64 |
| Russian | 0.64 | 0.64 | 0.65 | 0.63 |
| Filipino | 0.62 | 0.64 | 0.65 | 0.59 |
| Chinese | 0.40 | 0.42 | 0.41 | 0.39 |

for the Chinese was only slightly better than chance, and with an F1 distribution largely overlapping with chance level. Except for the Chinese sample, the model outperformed the baseline model (trained on randomly shuffled outcome labels) on all language samples. Results further indicated that the semantic features *pain*, *sad*, *fearful*, *unpleasant*, *disgusted* and *slow* were the most robust positive predictors of depression in the cross-linguistic model, while the features *number*, *LowerLimb*, *happy*, *taste*, *drive* and *benefit* were the most robust negative predictors of depression across bootstraps (Fig 4).

## Classification in individual language samples

One Decision Tree Ensemble model was trained and evaluated for each language separately, following the same procedure as described for the cross-linguistic model. Selected hyperparameters for each language model are reported in S1 Table. Similarly, robustness tests were performed with bootstrapping to obtain robust performance estimates and feature importance effects against random variations in the training data.

All models performed better than chance with the Filipino model performing best (F1 = 0.71, ROC-AUC = 0.73) and the German model performing worst (F1 = 0.56, ROC-AUC = 0.59) (Table 4). All models showed large discrepancies between performance on training data and test data, indicating some degree of overfitting.

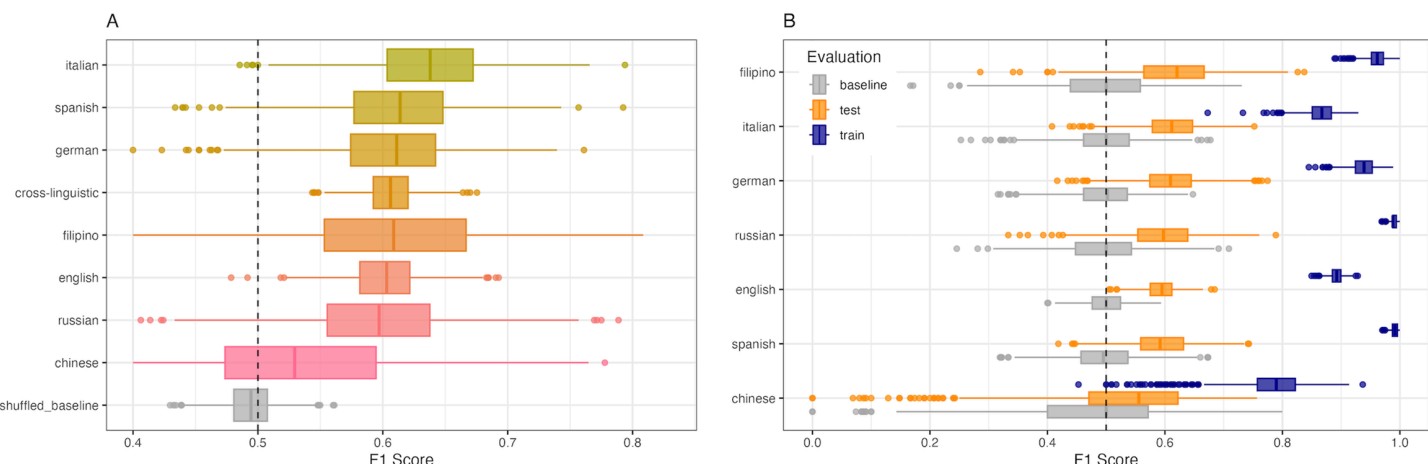

**Fig 3. Bootstrapped robustness tests all models.** Bootstrapped performance (F1) of the A) cross-linguistic model on the full sample (cross-linguistic) and each language-specific test sample and B) individual language-specific models.

**Fig 4. Cross-linguistic model semantic effects.** Bootstrapped SHAP values for the cross-lingusitic model. Positive SHAP values indicate that higher scores on this feature (more proximal demonstrative responses) drives predictions towards *depression* while negative SHAP values indicate that higher scores on this feature drives predictions towards *control*.

**Table 4. Performance of language-specific models.**

| Model | F1 | ROC-AUC | Precision | Recall |
|---|---|---|---|---|
| English | 0.63 | 0.61 | 0.59 | 0.67 |
| German | 0.56 | 0.59 | 0.60 | 0.53 |
| Italian | 0.59 | 0.64 | 0.67 | 0.53 |
| Spanish | 0.64 | 0.64 | 0.63 | 0.65 |
| Russian | 0.62 | 0.61 | 0.59 | 0.66 |
| Chinese | 0.67 | 0.69 | 0.73 | 0.61 |
| Filipino | 0.71 | 0.73 | 0.75 | 0.68 |

Bootstrapped robustness tests indicated that all models robustly outperformed the baseline model, except for the Chinese model for which the F1 distribution across bootstraps largely overlapped with that of the baseline model (Fig 3B). Across bootstraps the Filipino model performed best (median F1 = 0.62, SD = 0.08), followed by the Italian model (median F1 = 0.61, SD = 0.05) and the German model (median F1 = 0.61, SD = 0.05), the Russian model (median F1 = 0.60, SD = 0.07) and the English model (median F1 = 0.60, SD = 0.03) and the Spanish model (median F1 = 0.59, sd = 0.05). The Chinese model performed worst (median F1 = 0.56, SD = 0.12). F1-score distributions across bootstraps were wider (indicated by larger SDs) for languages represented by smaller samples than the larger language samples, indicating that performance in smaller samples were less robust to data sampling.

Pairwise rank correlations of SHAP values for the 65 semantic features between models showed that feature importance estimates of the cross-linguistic model was positively correlated with those of each separate language model, and most strongly with the Italian, Spanish and German models (Fig 5). Further, results indicated that the English, German, Spanish,

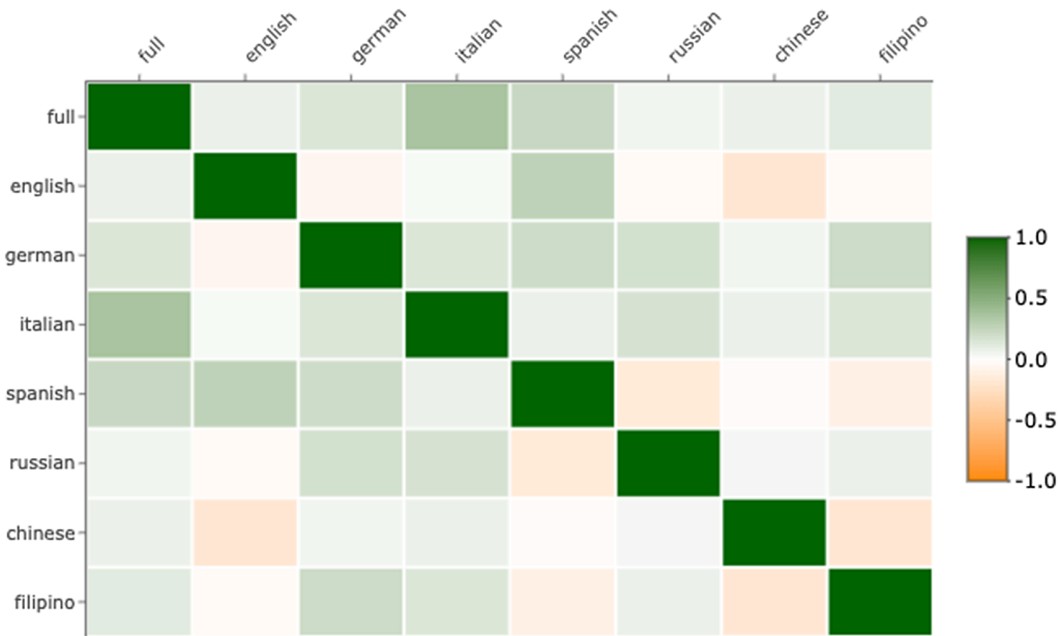

**Fig 5. Rank correlations of feature effects across models.** Pairwise rank correlation matrix of feature SHAP values across all models. Stronger correlations indicate a larger cross-linguistic correspondence between semantic features, indicative of depression.

and Italian models exhibited most similar feature effects, with the exception of German and English models exhibiting a weak negative rank correlation. The Russian and Filipino model showed strongest correlation in feature effects with the German and Italian model, and negative correlations with the Spanish and Chinese model. The Chinese model exhibited weak rank correlations with all other models, and negative correlations with the English and Filipino model. Fig 6 shows the averaged SHAP values of the semantic features across bootstraps for each model. These indicate that the features *pain*, *sad*, *fearful*, *unpleasant* and *disgusted* appear among the positive predictors of depression shared across most languages. In contrast, the features *happy*, *benefit*, *lowerLimb*, *near*, and *pleasant* appear as the negative predictors of depression shared across most languages. The most notable differences between languages is the negative effect of *dark* and the strong positive effect of *loud* observed in the Filipino model, as well as the strong positive effects of *happy* and *number* in the Russian model, which had a negative effect in the remaining models. For feature importance variability across bootstraps in each language model, see S2 Appendix.

## Discussion

Previous work has shown that the Demonstrative Choice Task elicits behaviors that can be used to infer self-reported depression in native English speakers. In an English DCT, demonstrative reference appears to reliably capture semantic effects of experiential states related to depression, such as high negative valence, for which individuals with high depression symptom scores are more likely to use the proximal demonstrative form, than the control group. The present work addressed the extent to which such effects are language specific, or whether they generalize across languages with different demonstrative systems and origin. Further, we investigated whether classification can be performed from DCT-based semantic profiles of participants, moving beyond item-level effects. Results suggested that simple lexical choices in the DCT capture semantic differences in experiential states related to depression, and may be used to map individuals along a broad semantic space, potentially providing novel insights into individual differences in disorder states and etiology.

The multi-language model, including all seven languages, reliably classified depression from semantic profiles of participants, and performance was robust against random variations in the training sample. Although stratification ensured equal distribution of each language in train and test sets, the between-language distributions was strongly imbalanced, with the English, German, Spanish and Italian largely over-represented. Thus, the good performance of the model could have been driven by high performance in one or more of these four languages. Evaluating the cross-linguistic model on each language-specific test sample indicated that it robustly performs better than the baseline model on all language samples, except for Chinese. However, performance was generally less robust to data sampling in smaller samples compared to larger samples. Further, the difference in classification performance of the cross-linguistic model across language samples was small, indicating that performance of the model was not mainly driven by patterns in the larger language samples. Rather, these results suggest that the model learned patterns associated with depression that are largely shared across languages. As discussed in detail further below, these seem to include more frequent use of the proximal demonstrative for semantic dimensions of the negative emotion domain and increased use of the distal demonstrative for dimensions of the positive emotion domain, compared to control subjects. This may reflect an increased psychological proximity of negative emotion domains accompanied by increased psychological distance of the positive emotion domains.

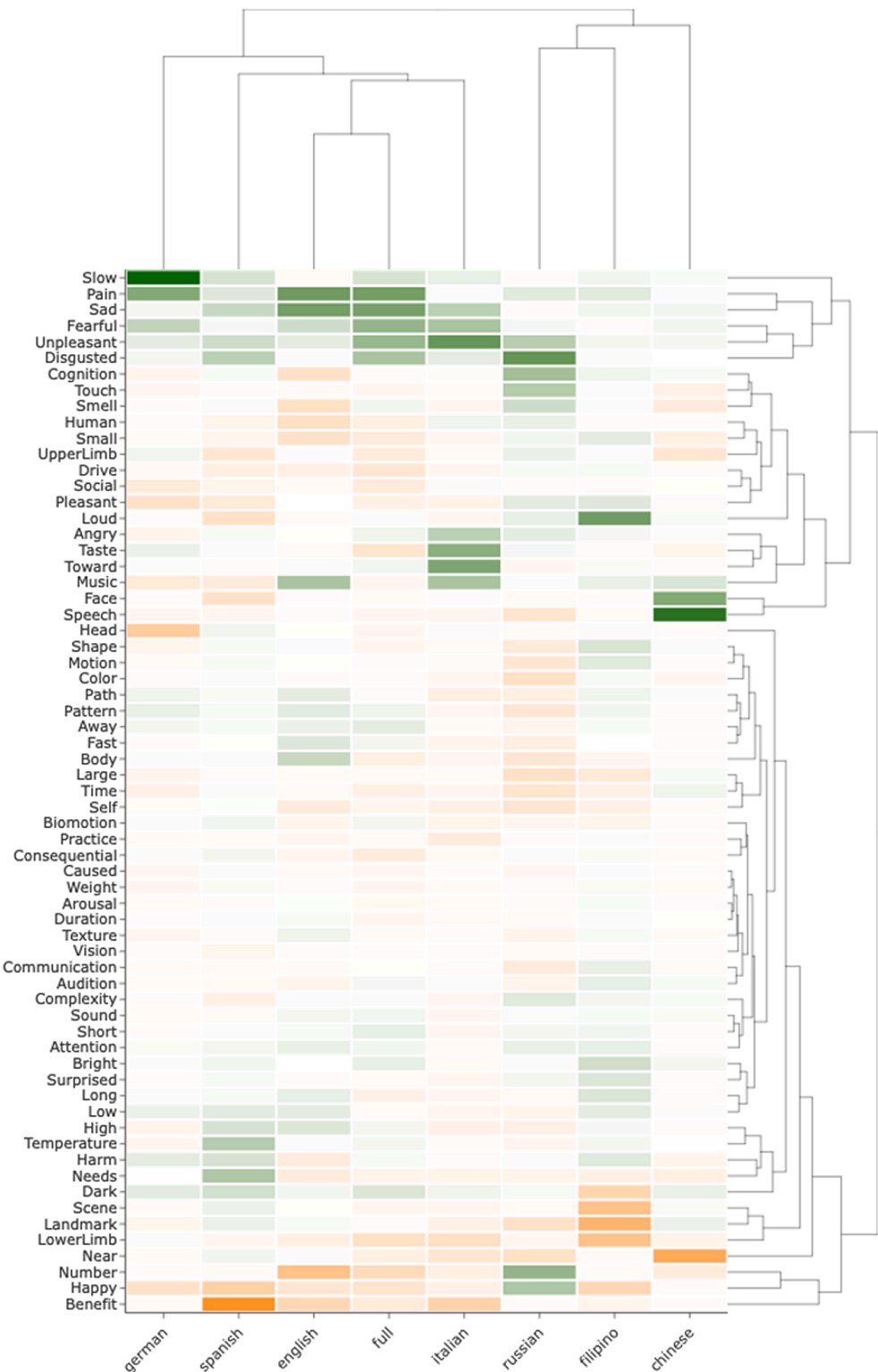

**Fig 6. Heatmap of semantic effects across models.** Mean SHAP scores of semantic features across bootstraps for each model. Mapped with hierarchical clustering. Green values indicate positive SHAP values. Orange indicate negative SHAP values.

Results of classification models trained individually for each language supported the inference that the relationship between semantic effects in DCT behavior and depression generalize across languages, again with the exception of Chinese (performance not better than chance). While all models robustly outperformed baseline models, performance estimates of smaller samples (Filipino and Russian) exhibited higher sensitivity to random variations in the data than those of larger samples (English, German, Spanish and Italian). Interestingly, classification performance did not seem to benefit particularly from language-specific training, for which performance estimates are highly similar to those obtained in the cross-linguistic model. However, these models are trained on substantially smaller samples, and robustness tests did show increased sensitivity to data sampling in smaller samples, and more outlier performance scores below chance level during bootstrapping. Thus, it is unclear whether language-specific training would increase performance of the language-specific models if trained on more sufficient sample sizes.

While most semantic features had small or zero importance for predictions across all models, the strongest positive and negative predictive features appeared to be highly similar across both the cross-linguistic and each individual language model. Semantic features of the *emotion* domain in the Binder list, particularly negatively valenced features as *pain*, *sad*, *unpleasant*, and *disgusted* were the positive predictive features shared across most models (higher frequency of proximal responses drives predictions toward depression), and consistently among the 10 most important features across models. Contrary, the features s *happy*, *benefit*, *lowerLimb*, *near*, and *pleasant* were among the negative predictive features shared across most models (higher frequency of proximal responses drives predictions toward control). Generally, the positive predictive effects (driving predictions towards depression) were stronger than the negative predictive effects, which is similar to what was observed in the previous DCT study in English [36]. This indicates that the strongest behavioral differences in demonstrative reference associated with depression are more related to the negative emotion dimensions than the positive emotion dimensions. Further, results indicated that the Russian, Chinese and Filipino models exhibited largest differences in feature effects compared to the four other languages, while the English, German, Spanish and Italian models exhibited larger similarity in semantic patterns. However, those three models were based on significantly smaller samples than the remaining models and associated with less robust classification performance. Additionally, bootstrapped SHAP values distributions overlapped with zero for almost all features in these three languages. Hence, while these results may reflect interesting semantic differences associated with depression in Russian, Filipino, and Chinese, it is uncertain to which degree these reflect differences in semantic representations and their relationship with depression that would generalize to larger samples. The observation that the cross-linguistic model generalizes well to unseen data in both Russian and Filipino indicates that the semantic patterns associated with depression may be shared to a larger extent than indicated by the language-specific models, and that sample sizes for these languages may have been too small for the model to learn these patterns. However, investigations of these effects in larger samples are needed to draw conclusions on this. The observed semantic effects of the present study strongly resemble those observed in previous work on English, where depression was classified from raw responses (PCs) [36]. This indicates that subject-wise semantic profiles aggregated across DCT trials captures the semantic differences in demonstrative reference observed in depression. Further, these recover the same semantic effects as has previously been associated with depression in classification models based on social media text [10,28,29,42,43], in cognitive processing of emotional information [8,30,44] on neural processing of emotional linguistic [6,7,31] and non-linguistic [45,46] information.

## Limitations

Importantly, the semantic features used in this work were based on a learned mapping between Glove semantic space and the Binder feature space. Both of these are based on English; the former derived from neural network models trained on large English corpora and the latter derived from human ratings on English words. Thus, the semantic scores of items in the DCT, and following subject-wise semantic scores, reflect the semantic features of those items in English. It is possible, however, that some concepts differ in their semantic content between languages, for instance, in cases where one concept can have multiple meanings in one language and not the other. Further, the semantic ratings of items reflect the English associations with the given term, and it is likely that such associations will be increasingly different with larger differences between languages. This may in part explain why Filipino and Chinese differ most in the observed semantic effects. Using aggregated semantic scores across all trials for each participant as predictors likely reduced the impact of such effects, however, it cannot be ruled out that there are important semantic differences that are unaccounted for in the models reported here. The present analyses of Spanish and Filipino are challenged by the fact that the semantic profiles of participants were based only on the proximal and distal responses. The semantic profiles were constructed to reflect scores on a proximal-distal axis $[-1, 1]$ for each semantic feature, averaged across all trials. Thus, including the medial responses in computation of semantic profiles as done here, requires a choice of the spatial interpretation of the medial demonstrative relative to this axis (is it closer to the proximal, closer to the distal, or not on the same axis?). To avoid potential spurious effects in comparison between languages, we confined the analysis to the binary distinction between the proximal and distal points. Supplementary analyses (S4 Supplementary Experimental Procedures) compared the model without the medial response (*-m*), to two models contrasting proximal vs. medial+distal (medial coded as $-1$, *md*) and contrasting proximal+medial vs. distal (medial coded as 1, *pm*). In the Spanish case, the *md*-model showed highly similar results as the *-m*-model (F1 = 0.58, respectively, rank correlation of feature SHAP scores = 0.64), while the *pm*-model showed reduced performance (F1 = 0.55) and elicited larger differences in semantic effects compared to the *-m*-model (SHAP value rank correlation = 0.42). In contrast, for the Filipino case, performance was reduced in the *md*-model (F1 = 0.56), while *pm*-model exhibited improved performance (F1 = 0.64) and elicited larger differences in semantic effects compared to the model without medial responses (SHAP score rank correlation = 0.49). This suggests that behavioral differences in demonstrative reference related to depression is mainly expressed in the contrast between proximal and medial/distal responses in Spanish, while in Filipino they are mainly expressed in the contrast between proximal/medial and distal responses. Additionally, results indicated that in the Spanish case, medial responses may be less systematically related to depression than the proximal and distal responses, as the model without medial responses performed best. Contrary, the medial responses might be more systematically related to depression in the Filipino case, where the model coding them as proximal performed best of the three. However, given the limited number of medial responses in the Filipino data, these interpretations are only speculative. Importantly, in each language, both a model with and without the medial responses robustly outperformed the baseline model.

Both the cross-linguistic and the Chinese-specific models performed at chance level on depression classification in Chinese, indicating no relationship between semantic effects in DCT behavior and depression symptom severity. This may be due to the very small percentage of participants assigned to the depression group in the Chinese sample (21.5%). Since the data is downsampled to balance depression and control cases in training, there may have been

too little information for the model to learn robust distinguishing features. The top predictive features in the Chinese model did resemble those of the remaining languages, albeit with much smaller and less robust effects, indicating that a similar pattern may exist. However, results could also suggest that demonstrative reference in the Chinese DCT does not capture differences related to depression. It may be that responses in the Chinese DCT is not systematically associated with semantic properties of the referents, or that there is substantial individual variation in depressive states, thus not yielding consistent semantic patterns. Another potential explanation is related to the highly tabood nature of depression in Chinese culture [47]. This may have caused less reliable PHQ-9 responses by Chinese participants, which could impair the relationship between symptom scores and DCT behavior. Analyses predicting demonstrative choice effects from semantic features, similar to what was done in [22], as well as depression classification in a larger sample with more positive cases, is needed to tease these effects apart. Generally, the results indicate both an overlap and differences in semantic effects between languages, which may point to interesting cultural differences in either semantic effects in demonstrative choices, semantic effects associated with depression, or both. These differences, however, must be further investigated in language samples of equal and adequate sizes to clarify their extent and nature.

## Implications and future directions

Demonstrative reference implies a proximal/distal distinction, in relation to a conceptual space [19], which in natural communication is defined by a range of contextual factors including physical, psychological and referent-intrinsic factors. It is thus hypothesized that when all contextual anchors are removed, as in the DCT, choice of demonstrative form reflects proximity to a conceptual space confined to the current psychological state. Here, specific concepts may be more or less salient or available, based on current psychological and emotional experience, reflected in experienced proximity relative to the psychological space. The present results showed that semantic differences in demonstrative reference across 290 nouns reliably captured individual differences in mental states associated with self-reported depression symptom severity across seven different languages. This suggests that, in absence of a guiding context, demonstrative reference may be used to make inferences on the semantic characteristics of current mental state and individual differences hereof. Similar semantic differences in produced text and speech associated with depression [48,49], may also reflect such increased proximity or salience of these concepts in psychological space. A large body of literature suggests that differences in language output and usage is likewise predictive of a range of other individual differences in mental state, such as personality traits [12,50–52] and anxiety [28,53,54]. Future work may address the extent to which such individual traits can be detected from demonstrative reference in the DCT as well. This could open interesting avenues to assess the semantic characteristics of different states, in an implicit manner, potentially revealing novel features of maladaptive psychological states, that are difficult to detect in self-reports. Further, this approach may provide insights into individual differences within groups (e.g., within individuals with depression), that are not captured in standard symptom ratings scales, but may contain important information on specific salient features of individual disorder states.

The results reported here, however, do show substantial individual variation in DCT behavior, requiring relatively large samples to infer robust group differences. Further, it is evident that most of the semantic features have no effect on depression classification across all languages, and may even impair model inference. It may be possible to fine-tune the item-selection in the DCT, to more reliably capture the effects of interest. This could be done by

removing items of irrelevant semantic features and include more items with low and high scores on the most important features, to obtain higher and more robust classification performance, which would facilitate more reliably inferences at the individual level. Such a procedure could allow fine-tuning of the paradigm specific to different maladaptive states, such as depression and anxiety, or to specific individual traits such as extroversion/introversion, neuroticism or psychoticism. Contrary, maintaining a wide semantic feature space could allow detection of individuals who deviate significantly on some semantic dimensions, not usually associated with group of interest.

## Acknowledgments

A great thank you to Svetlana Kuleshova, Ling Guang and Maiza Shine Fredensborg for helping with translations of the DCT paradigm into Russian, Chinese and Filipino.

## Ethics statement

The project was approved by the Institutional Review Board at Aarhus University. All participants provided written consent prior to participating in the study.

## Supporting information

**S1 Fig. Depression group distribution by language sample.**
(TIF)

**S2 Fig. Gender distribution by depression group and language sample.**
(TIF)

**S3 Fig. Demonstrative choice distribution (full sample).**
(TIF)

**S4 Fig. Demonstrative choice distribution by language sample.**
(TIF)

**S5 Fig. Demonstrative choice distribution by depression group.**
(TIF)

**S6 Fig. Demonstrative choice distribution by depression group and language sample.**
(TIF)

**S1 Table. Selected hyperparameters for each model.**
(TIFF)

**S1 Appendix. Semantic score distributions across participants for each language sample (Figs A–G).**
(PDF)

**S2 Appendix. Semantic feature effects (SHAP) for each individual language model (Figs A–G).**
(PDF)

**S1 Supplementary Experimental Procedures. Notes on translations of the DCT.**
(PDF)

**S2 Supplementary Experimental Procedures. Replication analysis: Depression classification from principal component representations of DCT responses.**
(PDF)

**S3 Supplementary Experimental Procedures. Binder scores extrapolation.**
(PDF)

**S4 Supplementary Experimental Procedures. Comparison of models with different coding schemes for the medial demonstrative (in Spanish and Filipino).**
(PDF)

## Author contributions

**Conceptualization:** Line Kruse, Roberta Rocca, Emanuela Todisco, Mikkel Wallentin.

**Data curation:** Line Kruse, Mikkel Wallentin.

**Formal analysis:** Line Kruse, Mikkel Wallentin.

**Funding acquisition:** Roberta Rocca, Mikkel Wallentin.

**Methodology:** Line Kruse, Roberta Rocca, Emanuela Todisco, Cordula Vesper, Peter Thestrup Waade, Mikkel Wallentin.

**Project administration:** Line Kruse, Mikkel Wallentin.

**Supervision:** Roberta Rocca, Mikkel Wallentin.

**Validation:** Line Kruse, Emanuela Todisco, Cordula Vesper, Peter Thestrup Waade, Mikkel Wallentin.

**Visualization:** Line Kruse.

**Writing – original draft:** Line Kruse.

**Writing – review & editing:** Line Kruse, Roberta Rocca, Emanuela Todisco, Cordula Vesper, Peter Thestrup Waade, Mikkel Wallentin.

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
