## [Decision Letter · Decision Letter 0]

27 Mar 2025

PMEN-D-24-00582

This and that in depression: cross-linguistic semantic effects

PLOS Mental Health

Dear Dr. Kruse,

Thank you for submitting your manuscript to PLOS Mental Health. After careful consideration, we feel that it has merit but does not fully meet PLOS Mental Health’s publication criteria as it currently stands. Therefore, we invite you to submit a revised version of the manuscript that addresses the points raised during the review process.

I have received reviews from two experts in the field, and prior to reading the reviews, I read the paper for an independent opinion. The reviewers and I agree that there is much to like about your manuscript. It is an interesting and thorough work. However, both reviewers highlight issues with the methodology and how it's reported - Reviewer 1 recommends more clarity on the translation process, and Reviewer 2 mentions possible confounds arising from data imbalance. Additionally, Reviewer 2 raises several issues about the structure and clarity of the methods and results reporting, and I agree with these observations.

We look forward to receiving your revised manuscript.

Kind regards,

Kongmeng Liew

Academic Editor

PLOS Mental Health

Journal Requirements:

1. We ask that a manuscript source file is provided at Revision. Please upload your manuscript file as a .doc, .docx, .rtf or .tex.

Additional Editor Comments (if provided):

Reviewers' comments:

Reviewer's Responses to Questions

**Comments to the Author**

1. Does this manuscript meet PLOS Mental Health’s publication criteria? Is the manuscript technically sound, and do the data support the conclusions? The manuscript must describe methodologically and ethically rigorous research with conclusions that are appropriately drawn based on the data presented.

Reviewer #1: Yes

Reviewer #2: Yes

2. Has the statistical analysis been performed appropriately and rigorously?

Reviewer #1: Yes

Reviewer #2: Yes

3. Have the authors made all data underlying the findings in their manuscript fully available (please refer to the Data Availability Statement at the start of the manuscript PDF file)?

Reviewer #1: Yes

Reviewer #2: Yes

4. Is the manuscript presented in an intelligible fashion and written in standard English?

Reviewer #1: Yes

Reviewer #2: Yes

5. Review Comments to the Author

Reviewer #1: A study explores the connection between language and depression across multiple languages. Participants from seven different language backgrounds (English, German, Spanish, Italian, Russian, Chinese, and Filipino) completed a Demonstrative Choice Task (DCT). The DCT measures preferences for proximal ("this") versus distal ("that") demonstratives when presented with various nouns. The study aimed to determine if patterns observed in English, where depressed individuals tend to associate negative words with "this," generalise across other languages and cultures. Models were trained to classify depression based on the participants' semantic profiles. The results indicate that, with the exception of Chinese, similar patterns exist, suggesting that basic lexical choices in the DCT reflect semantic differences in experiential states related to depression, although smaller sample sizes affect the robustness of these conclusions.

Major comments:

- Although the cross-linguistic validity of the proposed method is raised as a concern in the Limitation section, Lines 182-183 still justify that the Binder "is not language specific". As the authors acknowledge, the results instead suggest a cross-linguistic limitation of this method. These lines should be removed or modified.

- The translation process of the English DCT seems still ambiguous. Did the authors ask professional human translators? Was machine translation involved? Did the authors manually refer to dictionaries on their own? Though the Acknowledgments section mentions a potential translator for Chinese and Filipino, what the "help" exactly means is unclear. In addition to the current elaboration of different demonstrative systems, the detailed translation process should be explained in the S1 Supplementary Experimental Procedures. This process affects the quality of DCT translations and, ultimately, the discussions and interpretations of the results.

Minor comments:

- Some paragraphs are too long. Consider breaking up such paragraphs to improve readability.

- Tables 1, 2 and 5 are currently formated as figures. Convert them into formal tables.

- Typo -- “Filipnio” instead of “Filipino” (Line 464)

Reviewer #2: Dear authors,

thank you for your interesting work. I have a few questions that you might want to address in a revised version:

- For a better understanding, could you explain what "contextual anchors" are? I can imagine, but it would be great to have an example.

- I am not 100% clear about the experimental setup of the DCT: The participants get one word and then they have to associate it with either "this" or "that", is that correct? Also here, it would be great to see an example or a screenshot of he interface the participants were dealing with.

- Could you please give an example how participants were assigned Binder features exactly?

- I might be missing something, but why are determiners only added for German and Tagalog (Table 2)?

- Could you explain in a clearer way why you did not use the DCT responses directly for classification?

- The imbalance of language-specific data samples and the apparent instability of the models (which might also be due to the data imbalance) introduce confounding factors that are difficult to ignore. I think you did all you could do to mitigate these issues, but I am wondering if a much larger dataset size would give very different results.

- Since the data is very imbalanced and also because the diagnosed group is more relevant in this task, it would be more interesting to see the performance (e.g., F1) of the *positive* class, additionally to the scores across classes.

- The poor performance of the NN is not surprising since the data are probably too small to identify any meaningful features. Did you perform hyperparameter optimization?

- The results for Chinese were apparently worse than the baselines -- any idea why? Do you attribute this only to the different language families? I think there might also be a cultural effect (also for languages within Europe of course) which might be worth exploring.

- "suggests that the model learned patterns associated with depression that are largely shared across languages": it would be great to see an analysis of what these patterns are / might be

Typos etc.

- l. 32: "In this study, we address " (use present tense)

- l. 33: ".. in the Demonstrative Choice Task can be used ..." (no comma)

- ll. 110: "depression classification models were performed" -> do you mean "depression classification was conducted"?

- l. 164: "Group was coded .."  "Groups were coded as either 0 or 1"

- l. 173: "rating were obtained " (plural)

- l. 187: "outcome groups" (plural?)

- l. 282: "each model was trained" (singular)

- l. 313: "significantly classified depression group"  not sure I understand, do you mean sth like "classified members of the depression group correctly"?

- l. 346: missing bracket

- There are some problems with table references, see e.g., l. 145

- typo in Table 2, "jealousy"

- The figures in the appendix seem to have a rather low quality

- For some references, the year is missing (in the bibliography)

In summary, I think this study is sound work given the data scarcity. The paper is sometimes a bit hard to read since it includes many different models and evaluation criteria, but I think having more examples and the tables/figures next to the main text body would help a lot. However, trying to simplify each paragraph might be worth the effort to have a clearer structure and make it easier for the reader.

6. PLOS authors have the option to publish the peer review history of their article (what does this mean?). If published, this will include your full peer review and any attached files.

**Do you want your identity to be public for this peer review?** For information about this choice, including consent withdrawal, please see our Privacy Policy.

Reviewer #1: No

Reviewer #2: No

---

## [Decision Letter · Decision Letter 1]

29 Aug 2025

This and that in depression: cross-linguistic semantic effects

PMEN-D-24-00582R1

Dear Miss Kruse,

We are pleased to inform you that your manuscript 'This and that in depression: cross-linguistic semantic effects' has been provisionally accepted for publication in PLOS Mental Health.

Best regards,

Karli Montague-Cardoso

Staff Editor

PLOS Mental Health

Reviewer #1:

Reviewer Comments (if any, and for reference):

Reviewer's Responses to Questions

**Comments to the Author**

1. If the authors have adequately addressed your comments raised in a previous round of review and you feel that this manuscript is now acceptable for publication, you may indicate that here to bypass the “Comments to the Author” section, enter your conflict of interest statement in the “Confidential to Editor” section, and submit your "Accept" recommendation.

Reviewer #1: All comments have been addressed

2. Does this manuscript meet PLOS Mental Health’s publication criteria? Is the manuscript technically sound, and do the data support the conclusions? The manuscript must describe methodologically and ethically rigorous research with conclusions that are appropriately drawn based on the data presented.

Reviewer #1: Yes

3. Has the statistical analysis been performed appropriately and rigorously?

Reviewer #1: Yes

4. Have the authors made all data underlying the findings in their manuscript fully available (please refer to the Data Availability Statement at the start of the manuscript PDF file)?

Reviewer #1: Yes

5. Is the manuscript presented in an intelligible fashion and written in standard English?

Reviewer #1: Yes

6. Review Comments to the Author

Reviewer #1: All the comments I previously pointed out have been precisely addressed. The responses and explanations were clear and informative.

7. PLOS authors have the option to publish the peer review history of their article (what does this mean?). If published, this will include your full peer review and any attached files.

**Do you want your identity to be public for this peer review?** For information about this choice, including consent withdrawal, please see our Privacy Policy.

Reviewer #1: No
